# MULTI-BIN BATCHING
# FOR INCREASING LLM INFERENCE THROUGHPUT

## ABSTRACT

As large language models (LLMs) grow in popularity for their diverse capabilities, improving the efficiency of their inference systems has become increasingly critical. Batching requests during LLM inference increases throughput by allowing multiple requests to be processed in parallel, making better use of hardware resources such as GPUs. However, the autoregressive nature of LLMs presents a challenge: requests often have varying execution times, causing resource underutilization, as hardware must wait for the longest-running request in the batch to complete before moving to the next batch. We propose Multi-Bin Batching, a simple yet effective method that can *provably improve LLM inference throughput* by grouping requests with similar execution times into predetermined bins. We evaluate multi-bin batching on various settings, showing consistent throughput improvements compared to standard batching approaches.

## 1 INTRODUCTION

Large Language Model (LLM) inference systems are becoming increasingly popular due to their various abilities, such as text generation (Li et al., 2024), coding assistance (Chen et al., 2021), and question answering (Jiang et al., 2021). As the demand for LLM inference systems grows, so does the need to optimize their efficiency. Several techniques have been proposed to improve the efficiency of LLM inference systems, and *batched inference* (Sheng et al., 2023; Kwon et al., 2023; Jin et al., 2023) is one of the most promising techniques among them.

With batched inference, multiple requests are processed simultaneously, using the underlying hardware's parallelism to improve throughput. It can be seen in Figure 1a that generating 100 tokens for each request in an increasing batch size improves throughput. We measure throughput for the Phi-3.5 Mini Instruct model by prompting it with"once upon a time" in various batch sizes, generating 100 tokens per batch index on an NVIDIA A100 80G GPU. Throughput is calculated as total tokens generated across all indices divided by total generation time, using greedy sampling.

However, batched inference comes with some critical drawbacks. The execution time of each request depends linearly on the number of tokens generated. In standard batched inference systems, a computing unit remains locked until the entire batch is processed, meaning all requests in the batch must be completed before the system is released. This can result in underutilization of resources, offsetting some of the throughput gains achieved through parallelism in batched inference. Recent studies have proposed dispatching additional requests to the computing node before the current batch is fully processed. This approach, known as continuous batching (Yu et al., 2022), requires fine-grained control of hardware, which is not always feasible. In distributed or cloud-based environments, hardware control is typically abstracted or inaccessible, making it impossible to implement continuous batching.

Inspired by this, a natural question arises: can we achieve near-optimal throughput from batched inference without depending on fine-grained, hardware-level controls? Addressing this challenge is crucial for achieving high LLM inference throughput, particularly in such environments where continuously dispatching additional requests is not feasible.

We propose a novel approach for optimizing batched inference by binning requests based on their output lengths. Instead of placing all requests into a single queue, we create multiple "bins", each serving as a waiting area for requests with similar output lengths. Incoming requests are assigned

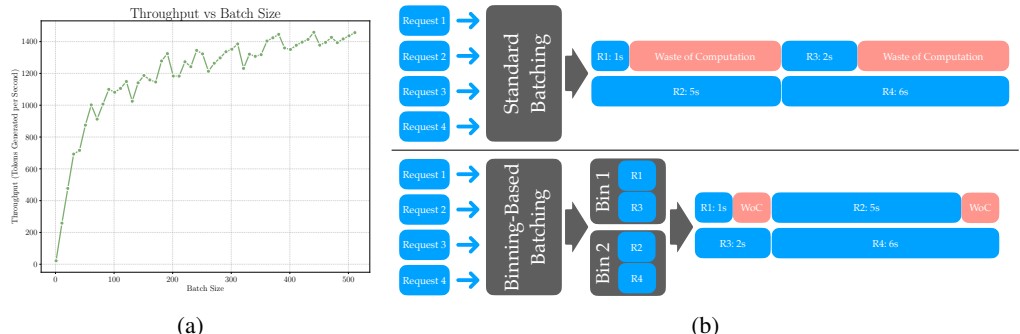

(a)                      (b)

Figure 1: (a) Batch serving improves the throughput for the LLM inference systems. (b) Standard batching causes under utilization of resources due to varying answer sizes.

to their corresponding bins based on these lengths, and batches are formed within each bin. Once a batch is ready, it is dispatched to a central queue to be processed.

Why is this approach beneficial? Consider the example illustrated in Figure 1b. Suppose four requests arrive at nearly the same time, with execution times of 1, 5, 2, and 6 seconds, respectively. In a standard batching system with a batch size of $B = 2$, the requests would be grouped based on their arrival time, forming two batches: (Request 1: *1s*, Request 2: *5s*) and (Request 3: *2s*, Request 4: *6s*). The total execution time would be 11 seconds (5 seconds for the first batch and 6 seconds for the second).

Now, consider our binning approach. Assume we have two bins: one for requests with output lengths between 1 to 3 seconds and another for those between 4 to 6 seconds. In this case, Request 1 and Request 3 would be placed in the first bin, while Requests 2 and 4 would go to the second bin. The resulting batches—(Request 1: *1s*, Request 3: *2s*) and (Request 2: *5s*, Request 4: *6s*)—would reduce the total execution time to 8 seconds (2 seconds for the first batch and 6 seconds for the second). This simple binning strategy demonstrates how aligning requests by output length can significantly improve LLM inference throughput.

Inspired by the toy example above, we propose multi-bin batching, a simple yet effective method that can *provably improve LLM inference throughput* by grouping requests with similar execution times into predetermined bins. We evaluate multi-bin batching on various settings using Microsoft's Phi-3.5-mini-instruct model on an Nvidia A100-80G GPU, demonstrating consistent throughput improvements compared to standard batching approaches. For instance, with the GSM8K dataset and an oracle output length estimator, multi-bin batching enhances throughput by up to 75% compared to standard batching systems. Our experiments span simulated results, and end-to-end LLM inference with oracle lengths, all showing significant performance gains as the number of bins increases.

To summarize, our contributions are as follows:

- We propose a novel binning-based batching system that can improve the throughput of LLM inference systems. Our batching system groups requests with similar execution times together based on predetermined bins.

- We use queueing-theoretical analysis to show that our multi-bin batching strategy can improve the throughput of LLM inference systems. We also show that how many bins are needed to achieve any desired throughput improvement.

- Our comprehensive experiments on real-world LLM models demonstrate that our proposed multi-bin batching system can enhance throughput by up to 75% compared to standard batching approaches.

## 2 RELATED WORK

**LLM Inference and Scheduling.**    Recent research has focused on optimizing large language model (LLM) inference through various scheduling techniques and tools from queueing theory. Wu et al. (2023) utilizes a novel preemptive scheduling algorithm, skip-join Multi-Level Feedback Queue, to improve the job completion time of LLM inference systems. Inoue (2021) considers a dynamic batching scenario (the system serves at most B jobs, if there are less than B jobs at the queue it serves them) and derives closed-form upper bounds for the mean latency. Cheng et al. (2024b) proposes a new scheduling method, slice-level scheduling that splits the maximum output length of the model into slices and serves batches slice by slice, which utilizes the memory more efficiently and reduces the response time. Llumnix (Sun et al., 2024) addresses the challenges of heterogeneous and unpredictable LLM inference requests through runtime rescheduling across multiple model instances, improving tail latencies and resource utilization. Yang et al. (2024) analyze LLM inference queueing delay using an M/G/1 model, demonstrating that enforcing maximum output token limits and optimizing batch size can significantly reduce latency.

**LLM Serving and Answer Length Estimation.**    There have been several studies on improving the throughput and the latency of LLM inference systems via estimating the answer length of the requests. Zheng et al. (2024) proposed a response time prediction model for LLM inference systems via prompting the model with an extra question to predict the response time. Instead of directly predicting execution times, Fu et al. (2024) predict the ranking of requests based on their execution times, and then propose a shortest-job-first scheduling algorithm to address the head-of-line blocking problem. Qiu et al. (2024) uses a light proxy model to predict the execution time of the requests and then uses a speculative shortest-job-first scheduling algorithm to improve the throughput of LLM inference systems. $S^3$ (Jin et al., 2023) estimates the answer length of the requests and uses it to optimize the memory efficiency of the LLM inference systems and it increases the effective batch size of the system thanks to the increased memory efficiency. Cheng et al. (2024a) uses input length to predict the response length of the requests and then uses it to optimize the batch size, it achieves higher throughput and reduces response time. Similarly, SyncIntellects (Lin et al., 2024b) enhanced response length prediction using a transformer-based model and implemented QoS-friendly length control, resulting in improved throughput and latency of LLM inference systems.

**LLM Inference Optimization.**    Recent studies have focused on optimizing the inference efficiency of LLM models through various techniques. Quantization has emerged as a key approach to reduce the memory footprint and improve the inference efficiency of LLM models. Methods like LLM.Int8() (Dettmers et al., 2022), GPTQ (Frantar et al., 2023), SmoothQuant (Xiao et al., 2023), and AWQ (Lin et al., 2024a) have demonstrated effective weight quantization techniques for LLM models, while QLoRA (Dettmers et al., 2024) combines quantization with parameter-efficient fine-tuning. Memory management innovations such as PagedAttention (Kwon et al., 2023) have significantly improved the serving throughput. KV cache optimizations, including compression techniques like Gear (Kang et al., 2024) have further improved the memory efficiency of LLM inference systems. Systems like FastServe (Wu et al., 2023), and FlexGen (Sheng et al., 2023) have integrated these techniques to create comprehensive LLM serving solutions.

## 3 PROBLEM SETUP AND THE MULTI-BIN BATCHING ALGORITHM

We begin by introducing the system model and key assumptions that form the foundation of our analysis. This model represents a typical LLM inference system as a queueing system with specific characteristics. Following this, we will propose our novel batching algorithm, which leverages a multi-binning approach to optimize request processing.

**Assumption 3.1.** *The LLM inference system is a single-server queueing system with an infinite queue length capacity. The system receives requests from a Poisson process with rate $\lambda$.*

Assumption 3.1 is a standard assumption in queueing theory, and it is well-suited for LLM inference systems since requests are typically generated by users in a random manner. The single server assumption is reasonable; it can further be easily extended to multi-server systems by assuming that the servers are identical and requests are served in a first-come-first-serve manner. In that case, the

effective arrival rate $\lambda$ can be divided by the number of servers, and one can derive similar results following the same analysis.

The system forms batches of size $B$ and serves them in a "first completed batch, first served" manner. The serving time of a batch of requests is the maximum of the serving times of the requests in the batch. This batching approach is suitable for LLM inference systems, as it enhances efficiency by optimizing the utilization of computing resources. The "first completed batch, first served" approach means that as soon as a batch is fully formed with $B$ requests, it becomes eligible for service, regardless of when its individual requests arrived. This allows for more efficient processing of requests, especially when combined with our batching strategy.

**Assumption 3.2.** *The service time of each request is independent and identically distributed (i.i.d.) with a uniform distribution in the range* $[l_{\min}, l_{\max}]$, *i.e.,* $l \sim \mathcal{U}(l_{\min}, l_{\max})$.

We make this assumption to simplify the analysis, and it is justified since LLM answer lengths typically fall within a specific range due to maximum token length limitations. We also extend our analysis to the case where the service time is exponentially distributed in the Appendix A.4.

In our theoretical analysis, we assume that the system always forms batches of size $B$ and then start processing them. However, in real systems, there could be a parameter that specifies the maximum time a batch can wait before it is processed. This way the system can ensure that the latency of a request does not exceed a certain threshold and the quality of service is maintained for all requests.

### 3.1 MULTI-BIN BATCHING ALGORITHM

We propose a novel batching algorithm that aims to improve the throughput of LLM inference systems. The key idea is to group requests into $k$ bins based on their service times before forming batches within each bin.

---

**Algorithm 1** Multi-Bin Batching with $k$-bins

---

**Require:** Decision boundaries for bins, $[l_{i-1}, l_i]$, $i = 1, \ldots, k$, batch size $B$, and serving policy
  1: **for** each incoming request **do**
  2:    Estimate its service time $l$
  3:    Assign the request to bin $i$ where $l_{i-1} \leq l < l_i$
  4: **end for**
  5: **for** each bin **do**
  6:    Form batches of size $B$ when available
  7:    Add completed batches to the service queue
  8: **end for**
  9: Serve batches from the service queue based on the serving policy provided
 10: The serving time of a batch is the maximum of the serving times of the requests in the batch

---

The multi-bin batching algorithm is illustrated in Figure 2. This algorithm works by first dividing the range of possible service times into $k$ bins. As requests arrive, their estimated service times are used to assign them to the appropriate bin. Within each bin, requests are grouped into batches of size $B$. As soon as a batch is com-

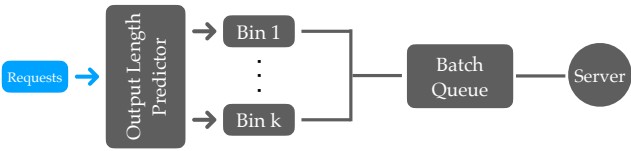

Figure 2: Multi-Bin Batching with $k$-bins

pleted in any bin, it is added to a service queue. The system then processes batches from the service queue in a "first completed batch, first served" order. This approach ensures that requests with similar service times are batched together, potentially reducing the overall serving time of each batch, while also allowing for efficient processing of completed batches across all bins.

## 4 THROUGHPUT ANALYSIS

In this section, we present a queueing-theoretical analysis to show that our multi-bin batching system can improve the throughput of LLM inference systems. To analyze the throughput of the system, we first derive the optimal decision boundaries for each bin in our batching system. Then, we derive the expected service time of a batch of requests with multi-bin batching algorithm.

The expected throughput of the system can be expressed as following proposition.

**Proposition 4.1.** *The expected throughput of the system is the ratio of the batch size to the expected service time. Specifically it can be written as,*

$$Throughput = \frac{B}{\mathbb{E}[t_{service}]}, \tag{1}$$

*where $B$ is the batch size and $\mathbb{E}[t_{service}]$ is the expected service time of a batch of $B$ requests.*

For our multi-bin batching system, we can derive the expected service time of a batch of $B$ requests as follows,

$$\mathbb{E}[t_{\text{service, k}}] = \sum_{i=1}^{k} \Pr(\text{bin} = i)\mathbb{E}\left[\max_{j \in [B]} x_j | \text{bin} = i\right], \tag{2}$$

where $\Pr(\text{bin} = i)$ is the probability that a batch served by the system is in bin $i$, and $\mathbb{E}\left[\max_{j \in [B]} x_j | \text{bin} = i\right]$ is the expected service time of a batch of $B$ requests from bin $i$. We also denote it as $\mathbb{E}[t_{\text{service, k}}]$ to emphasize that it is the expected service time of multi-bin batching system with $k$ bins. Then, the first step is to determine the decision boundaries for each bin in the multi-bin batching system for a fixed number of bins $k$. The following lemma provides the optimal decision boundary for each bin.

**Lemma 4.1.** *Under Assumption 3.2 and a fixed number of bins $k$, the throughput of the system is maximized when each bin has equal probability mass, and the decision boundaries are determined as follows,*

$$l_{i-1} = l_{\min} + \frac{i-1}{k}(l_{\max} - l_{\min}), \quad l_i = l_{\min} + \frac{i}{k}(l_{\max} - l_{\min}), \quad i \in [k]. \tag{3}$$

The proof of Lemma 4.1 is provided in the Appendix A.1. We first show that the expected service time is a convex function of the decision boundaries, and then we show that it is minimized when each bin has equal probability mass.

Given the optimal decision boundaries in Lemma 4.1, for a fixed number of bins $k$, we can have the following theorem for the expected throughput of the system.

**Theorem 4.2.** *Under Assumption 3.2, the expected throughput of the multi-bin batching with $k$ bins is,*

$$Throughput_k = \frac{B}{\mathbb{E}[t_{service, k}]} = \frac{B}{\frac{l_{\max}+l_{\min}}{2} + \frac{1}{k}\left(\frac{B}{B+1}l_{\max} + \frac{1}{B+1}l_{\min} - \frac{l_{\max}+l_{\min}}{2}\right)}, \tag{4}$$

*and it is an increasing function of the number of bins $k$.*

The proof of Theorem 4.2 is provided in the Appendix A.2. The proof utilizes the optimal decision boundaries in Lemma 4.1 to derive the expected service time of a batch of $B$ requests with $k$ bins. Then, we derive the expected throughput of the system with $k$ bins as a function of the number of bins $k$.

**Remark 4.1.** *The standard batching system is a special case of the multi-bin batching system with $k = 1$. If we substitute $k = 1$ into Equation 4, we can derive the expected throughput of the system with standard batching. Since the expected throughput of the system with the multi-bin batching is an increasing function of the number of bins $k$, the throughput of the system with our multi-bin batching system is higher than the standard batching system. Hence, the multi-bin batching system can improve the throughput of the system.*

**Remark 4.2.** *The expected throughput of the system with multi-bin batching is an increasing function of the number of bins $k$. As the number of bins $k$ goes the infinity, the expected throughput of*

*the system with multi-bin batching converges and we denote this as the maximum capacity of the system, which is,*

$$c_{\max} = \lim_{k \to \infty} Throughput_k = \frac{B}{\frac{l_{\max} + l_{\min}}{2}}. \tag{5}$$

*This convergence can be interpreted as follows: when $k$ becomes infinitely large, the bins become so fine-grained that the overall expected service time for the batch approaches the expected service time of a single request. However, we still process $B$ requests simultaneously, meaning that the throughput becomes $B$ times the single-server, non-batched throughput, which is $1/\mathbb{E}[T]$, where $\mathbb{E}[T]$ is the mean service time of a single request.*

In the following theorem, we derive the smallest integer $k$ that satisfies any given throughput less than the maximum capacity of the system.

**Theorem 4.3.** *Under Assumptions 3.1, and 3.2, for any $\epsilon > 0$, the desired throughput of the system $c_{\max} - \epsilon$ can be achieved by the multi-bin batching system with $k$ bins, where $k$ is the smallest integer satisfying the condition,*

$$k \geq \left\lceil \frac{(c_{\max} - \epsilon)\left(\frac{B}{B+1}l_{\max} + \frac{1}{B+1}l_{\min} - \frac{l_{\max}+l_{\min}}{2}\right)}{\epsilon \frac{l_{\max}+l_{\min}}{2}} \right\rceil = O\left(\frac{1}{\epsilon}\right). \tag{6}$$

The proof of Theorem 4.3 is provided in the Appendix A.3. The proof utilizes the expected throughput derivation in Theorem 4.2 to find the smallest integer $k$ that satisfies the desired throughput of the system.

Figure 3 shows the average throughput of the system with different number of bins $k$ as a function of the arrival rate $\lambda$. In this figure, we assumed that the batch size $B = 128$, the minimum service time $l_{\min} = 1$, and the maximum service time $l_{\max} = 20$. Therefore, the maximum capacity of the system is $c_{\max} = \frac{128}{\frac{20+1}{2}} \approx$ 12.3. We submit 128000 requests to the system and measure the time taken to process all the requests. We run the simulations for 10 times and report the average throughput of the system. Then, we calculate the average throughput of the system as the number of requests processed per unit time. It can be observed that the throughput of the system with multi-bin batching increases as the number of bins $k$ increases and when $k = 5$, the throughput of the system is close to the maximum capacity of the system. However, the binning idea comes with a trade-

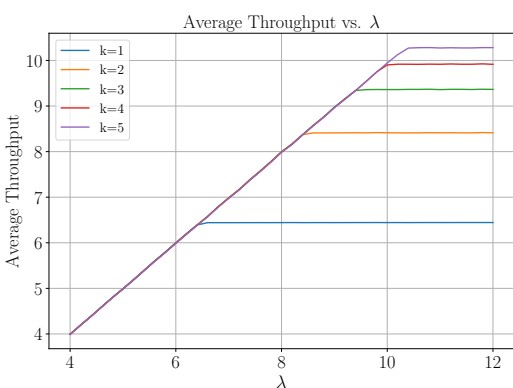

Figure 3: Average throughput of the system with multi-bin batching vs the arrival rate $\lambda$ for different number of bins $k$.

off, as the number of bins $k$ increases, the time to construct a batch of requests increases, which may lead to higher latency. In the next section, we evaluate the latency of the system with multi-bin batching and compare it with the standard batching system.

## 5 LATENCY ANALYSIS

We define the latency of a request as the time taken to complete a request from the time it is submitted to the system. The latency of a request consists of two components: the queuing time and the service time. In the previous section, we discussed the expected service time of a request for our multi-bin batching system. In this section, we analyze the queuing time of a request. The queuing time of a request is the time it spends waiting in the queue before it is processed. This time can also be decomposed into two components: the time spent waiting to complete the current batch and the time spent waiting for the current batch to start processing. In this analysis, we focus on the time spent waiting to complete the current batch, as it is the dominant component of the queuing

time in an underloaded system, which are common in cloud computing environments. In such systems, it is reasonable to assume that the time spent waiting for the current batch to complete remains the dominant factor in queuing time. This is because underloaded systems typically have shorter queues and less contention for resources, making the wait time between batches relatively insignificant compared to the time required to complete the batch itself. Therefore, we make the following simplifying assumption for the purpose of latency approximation:

**Assumption 5.1.** *The number of servers in the system is infinite. Therefore, whenever a batch is ready to be processed, it is immediately processed.*

Under this assumption, we can provide a lower bound on the latency of a request in our system because the time spent waiting for the current batch to start processing is negligible. We denote the expected latency of a request as $\mathbb{E}[t_{\text{latency}}]$. The following lemma provides the expected latency of a request in our system with the assumption of infinite servers.

**Lemma 5.1.** *Under Assumptions 3.2, and 5.1, and given the arrival rate $\lambda$ and $k$-bins with equal probability mass, the expected latency of a request is given by*

$$\mathbb{E}[t_{latency}] = \frac{l_{\max} + l_{\min}}{2} + \frac{1}{k}\left(\frac{B}{B+1}l_{\max} + \frac{1}{B+1}l_{\min} - \frac{l_{\max} + l_{\min}}{2}\right) + \frac{B-1}{2\lambda}k. \quad (7)$$

The proof of Lemma 5.1 is provided in Appendix B.1. The proof utilizes the fact that the arrival process is Poisson and the effective arrival rate of each bin is $\lambda/k$.

**Remark 5.1.** *The previous lemma provides the expected latency of a request in our system under the assumption of infinite servers. Therefore, it provides a lower bound on the latency of a request in our system with the assumption of one(or finite) server. Under the regime with low load factor, the assumption of infinite servers is reasonable and it provides a good approximation of the latency of a request in our system. However, as the load factor increases, the assumption of infinite servers becomes less accurate.*

In Figure 4, we plot the expected latency of a request as a function of the arrival rate $\lambda$ for different number of bins $k$. We use the parameters $l_{\min} = 1$, $l_{\max} = 20$, $B = 128$, and $k = 1, 2, 3$. We submit 128000 requests to the system and measure the latency of each request. We run the simulations for 10 times and report the average latency of a request. It can be seen that our Lemma 5.1 provides a good approximation of the latency of a request in our system when the arrival rate is low. As the arrival rate increases, the average latency of a request decreases until the arrival rate reaches the expected throughput of the system. Overall, our multi-bin

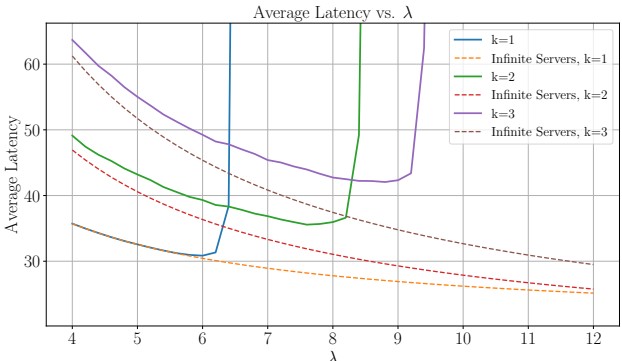

Figure 4: The expected latency of a request vs the arrival rate $\lambda$ for different number of bins $k$.

batching system with $k$-bins can provide a higher throughput compared to the standard batching system with a small increase in the latency of a request. In Appendix A.4, we provide the results for the case where the service time is exponentially distributed.

## 6 LLM EXPERIMENTS

To thoroughly analyze the throughput improvements from our multi-bin batching approach, we conduct two different experiments, with increasing levels of realism. Each experiment has two main components: the **service time** for a request and the **bin** to place the request into. In the first experiment, we model the service time as a linear function of the number of tokens generated by the model and use the known service time to predict the bin, referred to as oracle bin predictions. In the second experiment, we replace the linear model and instead send requests to a language model and

use the actual inference time. Across all experiments, we simulate requests arriving to our system as a Poisson process with rate $\lambda$.

## 6.1 SIMULATED RESULTS

To simulate the LLM inference time, we collect responses to questions from the GSM8K dataset (Cobbe et al., 2021) using Microsoft's Phi-3.5 Mini Instruct model (Abdin et al., 2024). We use greedy sampling on an Nvidia A100-80G GPU. We plot the number of generated tokens in a response against the inference time and perform a linear regression to approximate the time to generate each token. Given that the attention mechanism operates in a fully parallelizable manner for small input contexts, each token is processed efficiently, resulting in a constant time per output tokens and a linear relationship between the number of generated tokens and the overall inference time. This behavior holds true as long as the context size remains small, allowing parallel computation to maintain. This linear relationship can be seen at Figure 5.

Now, we simulate requests as questions from the GSM8K dataset according to an Poisson arrival process. We bin each request using the known responses lengths collected from Phi-3.5-mini-instruct. As described in the multi-bin batching algorithm, once a full batch size of $B$ requests is completed within a bin, that batch is added to a central queue. When a server is available, we simulate the service time for each request according to the linear model described earlier, and let the service time for the entire batch be the maximum of the individual requests' service time. This re-

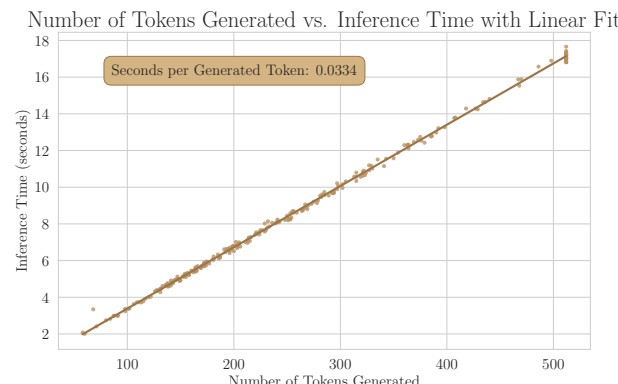

Figure 5: The linear relationship between the number of tokens generated and the inference time

flects the reality that a server is busy until the model has generated complete outputs for all requests within a batch.

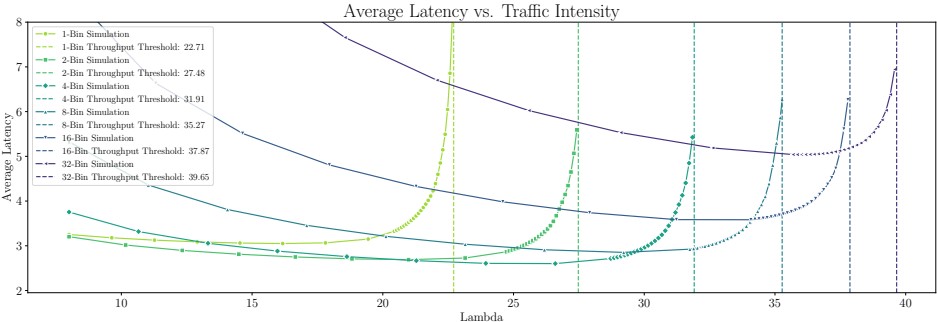

Figure 6: Throughput rises with more bins, while latency initially drops then climbs, illustrating the system's performance dynamics.

Figure 6 shows a comparison of latencies and maximum throughput across different numbers of bins as system traffic increases. Here, we fixed a batch size of 8 and simulate 8 servers, which can simultaneously serve batches. As the number of bins $k$ increases, so as does the throughput. Interestingly, when $k$ is small, such as $k \in [2, 4]$ the minimum latency is superior to the minimum latency without our method, when $k = 1$. This could be due to the difference between the output length and our assumption of uniform distribution.

## 6.2 END-TO-END LLM INFERENCE WITH ORACLE LENGTHS

To better understand the throughput gains from our method, we replace the linear model with actual time for an LLM to respond to a batch of requests. During this time the server is considered occupied. Specifically, we generate responses with Microsoft's instruction tuned Phi-3.5 Mini model, using a batch size of 8, a maximum of 1024 token, and a single simulated server, running on an Nvidia A100-80G.

Rather than simulate various arrival rates, we simulate a single large arrival rate, effectively equivalent to all requests arriving at once and record the throughput after completing all requests.

In this scenario, there is no time spent waiting for enough requests to arrive before a batch can be constructed to be processed. In other words, the server is fully utilized throughout the simulation; therefore, the throughput will be approximately the maximum possible throughput. Similar to the results in section 6.1, as the number of bins $k$ increases, the throughput also increases. The throughput increases approximately 70% from no binning to 32 binning.

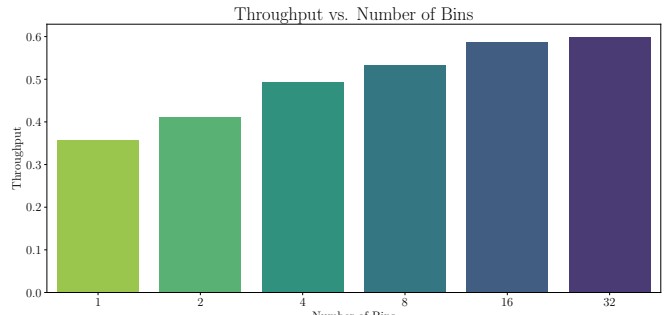

Figure 7: Inference throughput rises with more bins when output lengths are known.

## 7 CONCLUSION

This paper introduced multi-bin batching, a novel approach to optimize Large Language Model (LLM) inference systems without relying on fine-grained hardware controls. By grouping requests with similar output lengths, our method provides a provable throughput increase, mitigating resource underutilization in standard batched inference systems. Experiments demonstrated significant performance gains compared to standard batching. Our scalable solution contributes to LLM inference optimization and can be readily integrated into existing systems. As LLMs grow in importance, multi-bin batching enables more efficient deployments across various computing environments, especially where fine-grained hardware control is unfeasible. Future work could refine bin prediction models and explore adaptive binning strategies.

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

APPENDIX

## A  PROOFS FOR THROUGHPUT ANALYSIS

We give the proofs for the throughput analysis in this section.

### A.1  PROOF OF LEMMA 4.1

In this section, we provide the proof of Lemma 4.1.

*Proof.* We begin by defining the expected service time of a request in bin $i$ as follows,

$$\mathbb{E}\left[\max_{j\in[B]} x_j | \text{bin} = i\right] = \frac{B}{B+1}l_{i-1} + \frac{1}{B+1}l_i. \tag{8}$$

Since, the service time of a request in bin $i$ is uniformly distributed in the range $[l_{i-1}, l_i]$, the expected value of maximum of $B$ uniform random variables in the range $[l_{i-1}, l_i]$ is well-known and can be computed easily. Then, the expected service time of the system is given by,

$$\mathbb{E}\left[t_{\text{service, k}}\right] = \sum_{i=1}^{k} \Pr(\text{bin} = i)\mathbb{E}\left[\max_{j\in[B]} x_j | \text{bin} = i\right] \tag{9}$$

It can be written as,

$$\mathbb{E}\left[t_{\text{service, k}}\right] = \sum_{i=1}^{k} \frac{l_i - l_{i-1}}{l_{\max} - l_{\min}} \left( \frac{B}{B+1} l_{i-1} + \frac{1}{B+1} l_i \right). \tag{10}$$

For $k$ bins, we have $k-1$ decision boundaries, and $l_0 = l_{\min}$ and $l_k = l_{\max}$. We can denote the expected service time of the system as a function of $l_1, l_2, \ldots, l_{k-1}$ as follows,

$$f_k(l_1, l_2, \ldots, l_{k-1}) = \sum_{i=1}^{k} \frac{l_i - l_{i-1}}{l_{\max} - l_{\min}} \left( \frac{B}{B+1} l_{i-1} + \frac{1}{B+1} l_i \right). \tag{11}$$

We can compute the partial derivative of $f_k(l_1, l_2, \ldots, l_{k-1})$ with respect to $l_i, i \in [k-1]$ as follows,

$$\frac{\partial f_k(l_1, l_2, \ldots, l_{k-1})}{\partial l_i} = -\frac{B-1}{B+1} \frac{l_{i+1}}{l_{\max} - l_{\min}} + \frac{2(B-1)}{B+1} \frac{l_i}{l_{\max} - l_{\min}} - \frac{B-1}{B+1} \frac{l_{i-1}}{l_{\max} - l_{\min}}. \tag{12}$$

Then, the second-order partial derivative of $f_k(l_1, l_2, \ldots, l_{k-1})$ with respect to $l_i l_j, i, j \in [k-1]$ is given by,

$$\frac{\partial^2 f_k(l_1, l_2, \ldots, l_{k-1})}{\partial l_i \partial l_j} = \begin{cases} \frac{2(B-1)}{B+1} \frac{1}{l_{\max} - l_{\min}} & \text{if } i = j, \\ -\frac{B-1}{B+1} \frac{1}{l_{\max} - l_{\min}} & \text{if } |i-j| = 1, \\ 0 & \text{otherwise.} \end{cases} \tag{13}$$

The Hessian matrix of $f_k(l_1, l_2, \ldots, l_{k-1})$ is a tridiagonal matrix in the form of,

$$\nabla^2 f_k(l_1, l_2, \ldots, l_{k-1}) = \frac{B-1}{(B+1)(l_{\max} - l_{\min})} \begin{bmatrix} 2 & -1 & 0 & \cdots & 0 & 0 \\ -1 & 2 & -1 & \cdots & 0 & 0 \\ 0 & -1 & 2 & \cdots & 0 & 0 \\ \vdots & \vdots & \vdots & \ddots & \vdots & \vdots \\ 0 & 0 & 0 & \cdots & 2 & -1 \\ 0 & 0 & 0 & \cdots & -1 & 2 \end{bmatrix}. \tag{14}$$

The determinant of the Hessian matrix can be computed via the recursive formula for the determinant of a tridiagonal matrix as follows,

$$\det(\nabla^2 f_k(l_1, l_2, \ldots, l_{k-1})) = k \left( \frac{(B-1)}{(B+1)(l_{\max} - l_{\min})} \right)^{k-1} > 0 \tag{15}$$

Since $k > 1$ and $B > 1$, the determinant of the Hessian matrix is positive, which implies that the Hessian matrix is positive definite. Therefore, the function $f_k(l_1, l_2, \ldots, l_{k-1})$ is convex with respect to $l_1, l_2, \ldots, l_{k-1}$. Then, one can solve Equation equation 12 for $l_i$ by setting the partial derivative to zero, i.e., $\frac{\partial f_k(l_1, l_2, \ldots, l_{k-1})}{\partial l_i} = 0$. It can be seen that the optimal decision boundaries are given by,

$$l_i = l_{\min} + \frac{i}{k}(l_{\max} - l_{\min}) \quad \forall i \in [k-1]. \tag{16}$$

This completes the proof. $\qquad\square$

### A.2 PROOF OF THEOREM 4.2

In this section, we provide the proof of Theorem 4.2.

*Proof.* The expected service time of a batch of $B$ requests is,

$$\mathbb{E}[t_{\text{service, k}}] = \sum_{i=1}^{k} \Pr(\text{bin} = i) \mathbb{E}\left[ \max_{j \in [B]} x_j | \text{bin} = i \right] = \sum_{i=1}^{k} \frac{1}{k} \left( \frac{B}{B+1} l_i + \frac{1}{B+1} l_{i-1} \right), \tag{17}$$

because each bin has equal probability mass, and the service time of a batch of requests follows from the uniform distribution in the range $[l_{i-1}, l_i]$. If we substitute the optimal decision boundaries

in Equation 3 into Equation 17, we can derive the expected service time of a batch of $B$ requests with multi-bin batching as follows,

$$\mathbb{E}[t_{\text{service, k}}] = \frac{1}{k} \sum_{i=1}^{k} \frac{B}{B+1} \left( l_{\min} + \frac{i}{k}(l_{\max} - l_{\min}) \right) + \frac{1}{B+1} \left( l_{\min} + \frac{i-1}{k}(l_{\max} - l_{\min}) \right) \tag{18}$$

$$= \frac{1}{k} \frac{B}{B+1} \frac{k+1}{2}(l_{\max} - l_{\min}) + \frac{1}{k} \frac{1}{B+1} \frac{k-1}{2}(l_{\max} - l_{\min}) + l_{\min} \tag{19}$$

$$= \frac{l_{\max} + l_{\min}}{2} + \frac{1}{k} \left( \frac{B}{B+1} l_{\max} + \frac{1}{B+1} l_{\min} - \frac{l_{\max} + l_{\min}}{2} \right) \tag{20}$$

Then, the expected throughput of the system with multi-bin batching with $k$ bins is,

$$\text{Throughput}_k = \frac{B}{\mathbb{E}[t_{\text{service, k}}]} = \frac{B}{\frac{l_{\max}+l_{\min}}{2} + \frac{1}{k} \left( \frac{B}{B+1} l_{\max} + \frac{1}{B+1} l_{\min} - \frac{l_{\max}+l_{\min}}{2} \right)}, \tag{21}$$

and it can be observed that it is an increasing function of the number of bins $k$ since the denominator is decreasing with respect to $k$. $\qquad\square$

### A.3 Proof of Theorem 4.3

Here we provide the proof of Theorem 4.3.

*Proof.* The desired throughput of the system is $c_{\max} - \epsilon$. From Theorem 4.2, the expected throughput of the system with multi-bin batching with $k$ bins is,

$$\text{Throughput}_k = \frac{B}{\frac{l_{\max}+l_{\min}}{2} + \frac{1}{k} \left( \frac{B}{B+1} l_{\max} + \frac{1}{B+1} l_{\min} - \frac{l_{\max}+l_{\min}}{2} \right)}. \tag{22}$$

Then, we can find the smallest integer $k$ that satisfies the following condition,

$$c_{\max} - \epsilon \leq \text{Throughput}_k = \frac{B}{\frac{l_{\max}+l_{\min}}{2} + \frac{1}{k} \left( \frac{B}{B+1} l_{\max} + \frac{1}{B+1} l_{\min} - \frac{l_{\max}+l_{\min}}{2} \right)}. \tag{23}$$

We can solve the above inequality for $k$ to find the smallest integer $k$ that satisfies the desired throughput of the system,

$$(c_{\max} - \epsilon) \left[ \left( \frac{l_{\max} + l_{\min}}{2} \right) + \frac{1}{k} \left( \frac{B}{B+1} l_{\max} + \frac{1}{B+1} l_{\min} - \frac{l_{\max} + l_{\min}}{2} \right) \right] \leq B. \tag{24}$$

It can be simplified as follows,

$$B - \epsilon \left( \frac{l_{\max} + l_{\min}}{2} \right) + (c_{\max} - \epsilon) \frac{1}{k} \left( \frac{B}{B+1} l_{\max} + \frac{1}{B+1} l_{\min} - \frac{l_{\max} + l_{\min}}{2} \right) \leq B. \tag{25}$$

This implies,

$$k \geq \frac{(c_{\max} - \epsilon) \left( \frac{B}{B+1} l_{\max} + \frac{1}{B+1} l_{\min} - \frac{l_{\max}+l_{\min}}{2} \right)}{\epsilon \frac{l_{\max}+l_{\min}}{2}}. \tag{26}$$

Therefore, the smallest integer $k$ that satisfies the desired throughput of the system is given as in the statement of the theorem. $\qquad\square$

### A.4 Exponentially Distributed Service Time

In this section, we provide the expected service time of a batch of $B$ requests when the service time of each request is exponentially distributed with rate $\mu$. Hence, here we make the following assumption:

**Assumption A.1.** *The service time of each request is independent and identically distributed (i.i.d.) with an exponential distribution with rate $\mu$, i.e., $l \sim Exp(\mu)$.*

Then, for our multi-bin batching system, we need to decide the optimal decision boundaries to minimize the expected service time of a batch of $B$ requests. One can utilize the order statistics of the truncated exponential distribution (Joshi, 1978) to derive the expected service time of a batch of $B$ requests with $k$ bins. However, the exact values of truncated exponential order statistics are not easy to compute. Therefore, we use a simpler approach to derive an upper bound on the expected service time of a batch of $B$ requests with $k$ bins. For the bins before the last bin, we can upper bound the expected service time of a batch of $B$ requests as the decision boundary of that bin, i.e., for bin $i$, the expected service time of a batch of $B$ requests is upper bounded by $l_i$. For the last bin, the exact expected service time of a batch of $B$ requests is known and it is $l_{k-1} + \frac{H_B}{\mu}$, where $H_B$ is the $B$-th harmonic number. Then, we have the following lemma.

**Lemma A.1.** *Under Assumption A.1, the expected service time of a batch of $B$ requests with $k$ bins is upper bounded by,*

$$\mathbb{E}[t_{service, k}] \leq \sum_{i=1}^{k-1} \Pr(bin = i) l_i + \Pr(bin = k) \left( l_{k-1} + \frac{H_B}{\mu} \right). \tag{27}$$

*and this upper bound is minimized when the decision boundaries are set as,*

$$l_i = \frac{1}{\mu} \sum_{j=1}^{i} \log(L_{k-j}) \quad \forall i \in [k-1] \tag{28}$$

*where $L_m$ is defined recursively as:*

$$L_m = \begin{cases} H_B & \text{if } m = 1 \\ 1 + \log(L_{m-1}) & \text{if } m > 1 \end{cases} \tag{29}$$

*Proof.* The upper bound on the expected service time of a batch of $B$ requests with $k$ bins is derived based on the following observation.

$$\mathbb{E}[t_{\text{service, k}} | \text{bin} = i] \leq l_i \quad \forall i \in [k-1], \tag{30}$$

and the exact expected service time of a batch of $B$ requests with $k$ bins is given by $l_{k-1} + \frac{H_B}{\mu}$. This is well-known in the literature, it is the maximum of shifted exponential random variables. Then, the expected service time of a batch of $B$ requests with $k$ bins is upper bounded by the sum of the expected service time of each bin. The upper bound could be written as follows,

$$\mathbb{E}[t_{\text{service, k}}] \leq \sum_{i=1}^{k-1} \Pr(\text{bin} = i) l_i + \Pr(\text{bin} = k) \left( l_{k-1} + \frac{H_B}{\mu} \right). \tag{31}$$

The probability of each bin is given by $\Pr(\text{bin} = i) = \exp(-\mu l_{i-1}) - \exp(-\mu l_i)$. Then, we apply the following change of variables before minimizing the upper bound. Let $q_i = \exp(-\mu l_i)$ ($q_0 = 1$), then the upper bound can be written as,

$$\mathbb{E}[t_{\text{service, k}}] \leq \sum_{i=1}^{k-1} (q_{i-1} - q_i) \frac{\log(1/q_i)}{\mu} + q_{k-1} \left( \frac{\log(1/q_{k-1})}{\mu} + \frac{H_B}{\mu} \right) = f(q_1, q_2, \ldots, q_{k-1}). \tag{32}$$

It can be seen the upper bound function can be decomposed as a function of $q_1, q_2, \ldots, q_{k-1}$ and a multiplicative factor of $1/\mu$. Therefore, we will assume that $\mu = 1$ for simplicity. Then, we can write the upper bound function as,

$$f(q_1, q_2, \ldots, q_{k-1}) = \sum_{i=1}^{k-1} (q_{i-1} - q_i) \log(1/q_i) + q_{k-1} \left( \log(1/q_{k-1}) + H_B \right) \tag{33}$$

$$= \sum_{i=1}^{k-1} (q_i - q_{i-1}) \log(q_i) + q_{k-1} \left( H_B - \log(q_{k-1}) \right). \tag{34}$$

We can compute the partial derivative of $f(q_1, q_2, \ldots, q_{k-1})$ with respect to $q_i, i \in [k-2]$ as follows,

$$\frac{\partial f(q_1, q_2, \ldots, q_{k-1})}{\partial q_i} = \log(q_i) + \frac{q_i - q_{i-1}}{q_i} - \log(q_{i+1}) \quad \forall i \in [k-2]. \tag{35}$$

Then, the partial derivative with respect to $q_{k-1}$ is given by,

$$\frac{\partial f(q_1, q_2, \ldots, q_{k-1})}{\partial q_{k-1}} = H_B - \frac{q_{k-2}}{q_{k-1}}. \tag{36}$$

Then, the second-order partial derivative of $f(q_1, q_2, \ldots, q_{k-1})$ with respect to $q_i q_j, i, j \in [k-1]$ is given by,

$$\frac{\partial^2 f(q_1, q_2, \ldots, q_{k-1})}{\partial q_i \partial q_j} = \begin{cases} \frac{1}{q_i} + \frac{q_{i-1}}{q_i^2} & \text{if } i = j \text{ and } i \in [k-2], \\ \frac{q_{i-1}}{q_i^2} & \text{if } i = j = k-1, \\ -\frac{1}{q_{\max(i,j)}} & \text{if } |i-j| = 1, \\ 0 & \text{otherwise.} \end{cases} \tag{37}$$

Then, the Hessian matrix of $f(q_1, q_2, \ldots, q_{k-1})$ is a tridiagonal matrix in the form of,

$$\nabla^2 f(q_1, q_2, \ldots, q_{k-1}) = \begin{bmatrix} \frac{q_1+q_0}{q_1^2} & -\frac{1}{q_2} & 0 & \cdots & 0 & 0 \\ -\frac{1}{q_2} & \frac{q_2+q_1}{q_2^2} & -\frac{1}{q_2} & \cdots & 0 & 0 \\ 0 & -\frac{1}{q_3} & \frac{q_3+q_2}{q_3^2} & \cdots & 0 & 0 \\ \vdots & \vdots & \vdots & \ddots & \vdots & \vdots \\ 0 & 0 & 0 & \cdots & \frac{q_{k-1}+q_{k-2}}{q_{k-1}^2} & -\frac{1}{q_{k-1}} \\ 0 & 0 & 0 & \cdots & -\frac{1}{q_{k-1}} & \frac{q_{k-2}}{q_{k-1}^2} \end{bmatrix}. \tag{38}$$

The determinant of the matrix can be found using the following recursive formula:

$$f_n = A_{n,n} f_{n-1} - A_{n,n-1} A_{n-1,n} f_{n-2} \quad \forall n \in [2, k-1] \tag{39}$$

where $f_1 = A_{1,1}$ and $f_0 = 1$. For all $n \in [2, k-2]$, it can be seen that:

$$f_n = \frac{q_{n-1} + q_n}{q_n^2} f_{n-1} - \frac{1}{q_n^2} f_{n-2} \tag{40}$$

Our claim is that:

$$f_n = \frac{1}{q_1 q_2 \ldots q_{n-1} q_n^2} + \frac{1}{q_n} f_{n-1} \quad \forall n \in [2, k-2] \tag{41}$$

It holds for $n = 1$. Then, we can prove it by induction. Assume that it holds for $n-1$. Then, we can write the following:

$$f_n = \frac{q_{n-1} + q_n}{q_n^2} f_{n-1} - \frac{1}{q_n^2} f_{n-2} \tag{42}$$

$$= \frac{q_{n-1} + q_n}{q_n^2} \left( \frac{1}{q_1 q_2 \ldots q_{n-2} q_{n-1}^2} + \frac{1}{q_{n-1}} f_{n-2} \right) - \frac{1}{q_n^2} f_{n-2} \tag{43}$$

$$= \frac{1}{q_1 q_2 \ldots q_{n-1} q_n^2} + \frac{1}{q_n} f_{n-1} \tag{44}$$

Hence, it is proven by induction.

Then, we can compute the determinant of the Hessian as:

$$\det(\nabla^2 f) = f_{k-1} = \frac{q_{k-2}}{q_{k-1}^2} f_{k-2} - \frac{1}{q_{k-1}^2} f_{k-3} \tag{45}$$

We can replace the $f_{k-2}$ with the formula:

$$f_{k-2} = \frac{1}{q_1 q_2 \ldots q_{k-3} q_{k-2}^2} + \frac{1}{q_{k-2}} f_{k-3} \tag{46}$$

Then, we can compute the determinant of the matrix as:

$$\det(\nabla^2 f) = \frac{q_{k-2}}{q_{k-1}^2} \left( \frac{1}{q_1 q_2 \ldots q_{k-3} q_{k-2}^2} + \frac{1}{q_{k-2}} f_{k-3} \right) - \frac{1}{q_{k-1}^2} f_{k-3} \tag{47}$$

$$= \frac{1}{q_1 q_2 \ldots q_{k-2} q_{k-1}^2} \tag{48}$$

Therefore, the determinant of the Hessian matrix is as follows:

$$\det(\nabla^2 f) = \frac{1}{q_1 q_2 \ldots q_{k-2} q_{k-1}^2} > 0 \tag{49}$$

Since all $q_i$ are positive. Therefore, the upper bound for the total service time is a convex function of the decision points for the bins. Then, the optimal decision boundaries can be found by setting the partial derivative of the upper bound function to zero. We can start from the partial derivative with respect to $q_{k-1}$ as follows:

$$\frac{\partial f}{\partial q_{k-1}} = H_B - \frac{q_{k-2}}{q_{k-1}} = 0 \implies q_{k-2} = q_{k-1} H_B \tag{50}$$

Then, we can compute the partial derivative with respect to $q_i, i \in [k-2]$ as follows:

$$\frac{\partial f}{\partial q_i} = \log(q_i) + \frac{q_i - q_{i-1}}{q_i} - \log(q_{i+1}) = 0 \implies \frac{q_{i-1}}{q_i} = 1 + \log\left(\frac{q_i}{q_{i+1}}\right) \tag{51}$$

Utilizing the above equation and $q_0 = 1$, we can derive,

$$\frac{1}{q_1} = 1 + \log\left(\frac{q_1}{q_2}\right) \tag{52}$$

$$= 1 + \log\left(1 + \log\left(\frac{q_2}{q_3}\right)\right) \tag{53}$$

$$= 1 + \log\left(1 + \log\left(1 + \ldots + \log\left(\frac{q_{k-2}}{q_{k-1}}\right)\right)\right) \tag{54}$$

$$= 1 + \log\left(1 + \log\left(1 + \ldots + \log(H_B)\right)\right) = L_{k-1} \tag{55}$$

$$\implies q_1 = \frac{1}{L_{k-1}} \tag{56}$$

where $L_{k-1}$ is defined recursively as:

$$L_m = \begin{cases} H_B & \text{if } m = 1 \\ 1 + \log(L_{m-1}) & \text{if } m > 1 \end{cases} \tag{57}$$

Similarly, we can derive $q_i$ for $i \in [k-1]$ as follows:

$$q_i = \frac{1}{\prod_{j=1}^{i} L_{k-j}} \quad \forall i \in [k-1] \tag{58}$$

Then, the optimal decision boundaries are given by,

$$l_i = -\frac{1}{\mu} \log(q_i) = \frac{1}{\mu} \sum_{j=1}^{i} \log(L_{k-j}) \quad \forall i \in [k-1]. \tag{59}$$

This completes the proof. $\square$

Given the optimal decision boundaries in Lemma A.1, we can derive the expected service time of a batch of $B$ requests with $k$ bins.

**Corollary A.1.1.** *Under Assumption A.1, and the optimal decision boundaries in Lemma A.1, the expected service time of a batch of $B$ requests with $k$ bins is given by,*

$$Throughput_k = \frac{B}{\mathbb{E}[t_{service, k}]}$$

$$\geq \frac{B\mu}{\sum_{i=1}^{k-1} \frac{L_{k-i-1}}{\prod_{j=1}^{i} L_{k-j}} \cdot \sum_{j=1}^{i} \log(L_{k-1-j}) + \frac{1}{\prod_{j=1}^{k-1} L_{k-j}} \left(\sum_{j=1}^{k-1} \log(L_{k-1-j}) + H_B\right)} \tag{60}$$

*Proof.* The proof of corollary follows from the optimal decision boundaries in Lemma A.1 and the expected service time of a batch of $B$ requests with $k$ bins. $\square$

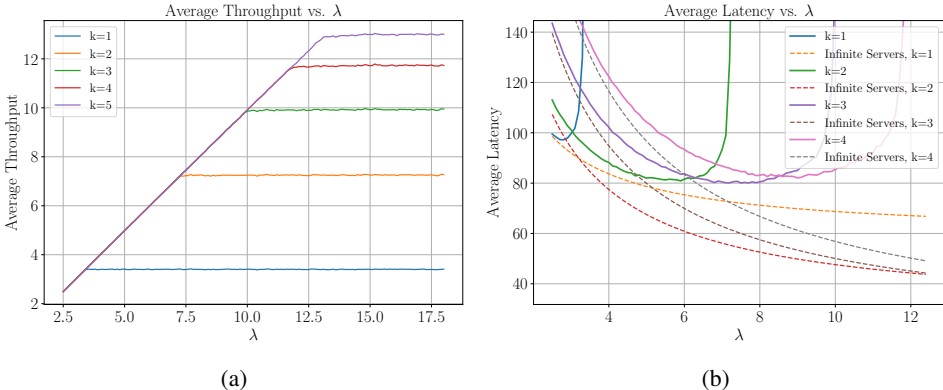

(a)                                                    (b)

Figure 8: (a) Throughput of the system with respect to the arrival rate $\lambda$ for different values of $k$. (b) Expected latency of a request with respect to the arrival rate $\lambda$ for different values of $k$.

Similary to the uniform distribution case, we provide numerical results for the exponentially distributed service time case. In Figure 8, we provide the throughput and expected latency of the system with respect to the arrival rate $\lambda$ for different values of $k$. We assume that the service time of each request is exponentially distributed with rate $\mu = 0.1$, the batch size is $B = 200$, and the total number of requests is $N = 200000$. We run the simulations for 10 different seeds and provide the average throughput and expected latency of the system. It can be observed that the throughput of the system increases with the number of bins $k$ with the multi-bin batching policy in Figure 8a. The average latency of the system depicted in Figure 8b decreases with the number of bins $k$. It can be seen that with the increasing number of bins, the system first achieves a lower latency but after a certain point, the latency starts to increase. This is different from the results in the uniform distribution case, where the latency increases with the number of bins.

# B    PROOFS FOR LATENCY ANALYSIS

## B.1    PROOF OF LEMMA 5.1

In this section, we provide the proof of Lemma 5.1.

*Proof.* Under the assumption of infinite servers, the latency consists of the time spent waiting to complete the current batch and the service time. Therefore, the expected latency of a request is given by

$$\mathbb{E}[t_{\text{latency}}] = \mathbb{E}[t_{\text{batch}}] + \mathbb{E}[t_{\text{service}}]. \tag{61}$$

The expected time spent waiting to complete the current batch is given by

$$\mathbb{E}[t_{\text{batch}}] = \sum_{i=1}^{k} \mathbb{P}(\text{bin} = i)\mathbb{E}[t_{\text{batch}}|\text{bin} = i]. \tag{62}$$

Since the bins are equally likely, the arrival rate for each bin is $\lambda/k$. Then, for each request in the batch, the expected time spent waiting to complete the current batch is given by

$$\mathbb{E}[t_{\text{batch}}|\text{bin} = i] = \frac{1}{B}\sum_{j=1}^{B}\frac{(B-j)k}{\lambda} = \frac{B-1}{2\lambda}k. \tag{63}$$

Then, the expected time spent waiting to complete the current batch is given by

$$\mathbb{E}[t_{\text{batch}}] = \frac{B-1}{2\lambda}k. \tag{64}$$

The expected service time of a request is given by Theorem 4.2. Therefore, the expected latency of a request can be derived as in the statement of the lemma.                                              $\square$