# OpenReview forum: "Multi-Bin Batching for Increasing LLM Inference Throughput"
_ICLR.cc/2025/Conference — ICLR 2025 Conference Withdrawn Submission_

### Official Review · Reviewer_cuTe · 2024-10-29

**Soundness:** 1
**Presentation:** 2
**Contribution:** 1
**Rating:** 3
**Confidence:** 4

**Summary:**

This paper proposes multi-bin batching, a method to enhance the throughput of LLM inference systems. It provides an analysis to quantify the throughput gains achieved by this approach and develops strategies to balance these gains with the additional latency introduced. Experimental results show that multi-bin batching improves LLM inference system throughput by up to 70% compared to standard inference.

**Strengths:**

1. This paper conducts a theoretical analysis, demonstrating the proposed batching method's correctness, effectiveness, and limitations.
2. It provides insights into achieving responsive and efficient LLM inference serving by incorporating answer length estimation.

**Weaknesses:**

1. The experiments in this paper are limited and do not fully support the claims of "various settings" (L87) or "comprehensive experiments" (L103). The experiments are confined to a single model, a single device, and a fixed batch size, which limits the ability to demonstrate the robustness and generalization of the approach across diverse scenarios.
 The baseline for comparison is restricted to standard batching inference, without evaluating the proposed approach against existing methods that improve LLM inference throughput by batching, such as those mentioned in the related works (e.g., [1]). Additionally, the details of experiment 6.2 are ambiguous, lacking clear information about the requests' sources, attributes (e.g., prompt length, output length), or how inference times are estimated.

2. The experiment in Section 6.2 is limited to a simulated environment rather than a real-world LLM inference system and is performed under the strong assumption that all requests arrive at once. This assumption has not been validated through providing sufficient evidence to demonstrate most real-world scenarios satisfy this requirement. As a result, the exploration of how latency from waiting for requests impacts response times is constrained to theoretical analysis.

3. The novelty of the proposed approach is limited by insufficient differentiation from prior work. The paper does not clearly explain the differences or relationships between this method and existing approaches that leverage batching, weakening its contribution. For instance, the paper claims that continuous batching is unsuitable for distributed or cloud-based environments (L45) while the proposed method works in these scenarios, yet no experiments are conducted in such environments to substantiate this claim, nor is the proposed method compared against approaches using continuous batching.

[1] Ke Cheng, Wen Hu, Zhi Wang, Hongen Peng, Jianguo Li, and Sheng Zhang. Slice-level scheduling for high throughput and load balanced llm serving. arXiv preprint arXiv:2406.13511, 2024b

**Questions:**

Some assumptions in the paper need further validation and discussion.
1. Are there any real-world examples or empirical analyses supporting the assumption in Section 3.2 (L172) that service times of different requests are uniformly distributed?
2. Are the proposed methods able to remain effective when the estimated answer lengths are inaccurate?

---

### Official Review · Reviewer_pt1B · 2024-10-30

**Soundness:** 3
**Presentation:** 4
**Contribution:** 3
**Rating:** 5
**Confidence:** 4

**Summary:**

This paper introduces a new queuing theoretic based method to batch LLM requests. In reality the method can be used for other ML workloads too as long as there is significant variability in the service time. The idea is to batch similarly sized requests (or at least have batches with close service time requirements). Once a batch is ready, it should be dispatched to run on the GPU. The authors proof the limits of the throughput and the decision boundaries of their system.

**Strengths:**

1. Thank you for submitting your work to ICLR. The paper is very well written generally.
2. The paper has a solid queuing theoretic models for the performance of the proposal system.

**Weaknesses:**

I truly enjoyed reading the paper. However, I could not help but wonder on a few more practical questions:
1. When you do the latency derivation, you assume you are operating at the low utilization regime. However, in your simulations, e.g., in Fig.4 you show results mostly on the high utilization regime (as \lambda increases). You also derive a model when the system is at maximum throughput. I am afraid that as you basically get into the higher regimes, the assumptions do not hold and all of a sudden your remark 5.1 is the truth. In addition, in fact, unless your average utilization is extremely low, queuing delays do occur at the low-utilization regime. For an M/M/1 queue, the theoretical average waiting time is actually ρ/(μ − λ) for a single queue system. So I am afraid that the latency in Figures 4 and 6 is a gross underestimate for higher  λ which will lead to also higher ρ. I think what would be interesting is to study when: the k=n queuing delays+bin-filling delays < k=1 queuing delays
The queuing delays for the k=n is probably lower than those k=1, but this will depend on the distribution of the sizes of the arriving tasks

2. Looking at eq.5, one main issue I see is that you really need large batch sizes for this to pay off. In addition, l_max and l_min will highly influence the perofrmance. There is an added complexity with multiple bins to manage and there is a huge risk of starvation which can really weaken the results. If you basically have dynamic batching to make sure that starvation does not happen, then there is a risk that your waiting time before you decide to schedule the tasks anyways even with no similarly sized batch will be extremely hard to tune, and will introduce worse tails response times.

3. How small is small for the context sizes for your simulated results to hold?

4.  The simulated results in Fig. 6 are a bit odd in my opinion. I am not really sure how the latency for larger k can be lower than k=1 at low λ. Even from the theory point-of-view, your queuing theory models do not support such a result as in essence it says that the time you need to wait to fill the bins does not matter. I think I would love to see more analysis on why is this happening.

**Questions:**

1. How will operating at high utilizations affect your results?
2.How will adding dynamic batching affect your result?
3.How small does the context size need to be for the simulation results to hold?
4. Why are Figures 4 and 6 much different?

I will be happy to change my scores based on the discussions.

---

### Official Review · Reviewer_GbZr · 2024-11-01

**Soundness:** 3
**Presentation:** 2
**Contribution:** 1
**Rating:** 3
**Confidence:** 4

**Summary:**

This paper proposes multi-bin batching for optimizing LLM serving systems, which models the inference system as a server queuing system and derives various performance guarantees. The proposed method groups requests with similar output lengths to improve inference throughput while mitigating resource under-utilizations in standard LLM serving systems. Both simulations and experiments on real testbeds are conducted to evaluate the proposed approach, which shows a throughput increase compared to standard LLM inference batching.

**Strengths:**

1 - The paper is well-written.

2 - Extensive theoretical analysis is provided.

3 - Both simulations and evaluations on testbeds are conducted.

**Weaknesses:**

1 - The proposed approach has a significant gap: how do you predict the output lengths or prompt service time? I can only find the term "Output Length Predictor" in Figure 2, but I have no idea how you implemented this module. Note that this is actually a significant challenge that most of the works in Paragraph 2, Section 2 focused on.

2 - In the introduction, the paper mentioned that "continuous batching requires fine-grained control of hardware, which is not always feasible." This assumption is somewhat ambiguous, as continuous batching can be supported by various modern and common hardwares. Specifically, vLLM has a lot of quick examples of deploying in cloud-based environments.

3 - Generally, I understand the idea that the paper models the LLM serving system as a server queuing system. However, the theory or technique behind such modeling is not new or novel. Once again, the more challenging problem here may be the output length prediction or service time estimation, as related works pointed out [1]. If the LLm serving system has high-confident output length predictions, optimizing the prompt batching would be relatively easy. How would the proposed method differ from existing classic server queuing systems without addressing the output length prediction problem?

4 - The evaluation is somewhat naive. Most of the experiments are conducted using simulations, as the only experiments on real testbeds are provided in Section 6.2. The evaluation simply uses a burst of requests for measuring the throughput. It would be better to have more realistic evaluations on the testbeds, such as using the real-world LLM inference traces [2] and presenting more metrics (e.g., latency and resource utilization).

5 - The evaluation lacks baseline comparison. A set of batching optimization works is mentioned in Section 2, yet the paper didn't compare the multi-bin batching with any of the state-of-the-art/practice solutions.

[1] Fu, Yichao, et al. "Efficient LLM Scheduling by Learning to Rank." arXiv preprint arXiv:2408.15792 (2024).

[2] Wang, Yuxin, et al. "Towards Efficient and Reliable LLM Serving: A Real-World Workload Study." arXiv preprint arXiv:2401.17644 (2024).

**Questions:**

Please see questions in weaknesses.

---

### Official Review · Reviewer_Ufwp · 2024-11-03

**Soundness:** 2
**Presentation:** 3
**Contribution:** 2
**Rating:** 3
**Confidence:** 4

**Summary:**

Batching requests is a common practice in LLM serving to improve GPU utilization and token throughput.
However, standard batched inference still leads to under-utilization since the completion of a batch depends on the longest ongoing request in the batch.
To resolve this issue, this paper proposes multi-bin batching, which places requests in multiple bins depending on their predicted length, and then executes batches in each bin based on scheduling policy.
The paper uses queueing-theoretical analysis to show that a multi-bin batching algorithm could effectively improve the throughput of the system.
Experiments based on ideal and real-world LLM models demonstrate the improvement of throughput compared to standard batching approaches.

**Strengths:**

1. The proposed algorithm is simple and the paper is easy to follow.
2. The paper provides a good and comprehensive theoretical analysis of the algorithm.

**Weaknesses:**

1. Some assumptions made in the paper seem to be far from realistic scenario.
2. Lack of evaluation on more realistic use cases of multi-bin batching algorithm.
3. Lack of sensitivity analysis on the effects of accuracy of length prediction.

**Questions:**

Thank you for submitting the paper to ICLR 2025!
I think this paper tries to tackle the important problem of improving GPU utilization and throughput of LLM serving.
The paper is well-written and easy to follow.
However, I do think several major drawbacks exist that may require major revisions to improve and better justify the paper, which I will explain below.

First, I am a bit confused on why continuous batching is difficult to achieve in cloud environments.
The paper says continuous batching requires fine-grained control of hardware, which is not always feasible.
However, according to the Orca paper, the core component that enables continuous batching should be an iteration-level request scheduler that is implemented in the LLM serving backend.
I believe modifications on the kernel level are not necessary to enable continuous batching.
The Orca paper also does not mention any specific hardware requirements.

I understand that assumptions made in the paper would better help derive the theoretical bounds for the algorithm, but I do think some assumptions are questionable.
For example, in Section 5 Latency Analysis, why is an underloaded system common in cloud computing environments?
My understanding is that in a throughput-oriented LLM serving scenario, it should be assumed that the system is loaded with requests continuously coming into the system.
In fact, the end-to-end LLM experiment in Section 6.2 assumes all requests are arriving at once at the beginning, which effectively simulates an overloaded system.

In addition, from Sections 6.1 and 6.2, it seems that both experiments assume the output length of the request is known.
Does the paper assume that the oracle output length predictor always outputs perfect prediction?
To make the experiment more realistic, I think the paper should consider the effectiveness of accuracy of prediction on the multi-bin batching algorithm.
This could be done by integrating existing length predictors such as S^3 or SyncIntellects.

In a realistic LLM serving scenario, it is also important to consider not only the throughput but the goodput of the system (i.e. how many requests can be completed with their SLOs satisfied).
In the multi-bin batching algorithm, the allocation of requests is independent to the request arrival time and SLO.
This means that in one bin, there may be requests that have entirely different arrival time and SLOs.
If swapping requests between batches is not possible, how could the scheduler guarantee a certain level of goodput of the system can be achieved?

Other questions:
1. In Algorithm 1, it says the batches from the service queue is served based on the serving policy provided. In the experiments, what is the serving policy used? Is it first-come-first-serve?
2. In Figure 4, how many servers are there for the solid lines?
3. What does the throughput threshold mean in Figure 6?
4. For the experiment in Section 6.2, are there any specific reasons to prefer a single large arrival rate over various arrival rates?

---

### Note · Authors · 2024-11-13

**Comment:**

We thank the reviewers for their time and feedback on our paper. After careful consideration, we have decided to withdraw our submission to develop further the work in alignment with the feedback provided.

**Withdrawal Confirmation:**

I have read and agree with the venue's withdrawal policy on behalf of myself and my co-authors.